# Rutin-Loaded Poloxamer 407-Based Hydrogels for In Situ Administration: Stability Profiles and Rheological Properties

**DOI:** 10.3390/nano10061069

**Published:** 2020-05-31

**Authors:** Elena Giuliano, Donatella Paolino, Maria Chiara Cristiano, Massimo Fresta, Donato Cosco

**Affiliations:** 1Department of Health Sciences, University “Magna Græcia” of Catanzaro, Campus Universitario “S. Venuta”, Viale S. Venuta, I-88100 Catanzaro, Italy; elena.giuliano@unicz.it (E.G.); fresta@unicz.it (M.F.); 2Department of Experimental and Clinical Medicine, University “Magna Græcia” of Catanzaro, Campus Universitario “S. Venuta”, Viale S. Venuta, I-88100 Catanzaro, Italy; paolino@unicz.it (D.P.); mchiara.cristiano@unicz.it (M.C.C.)

**Keywords:** hydrogels, poloxamer 407, rheology, rutin, microrheology, Turbiscan

## Abstract

Rutin is a flavone glycoside contained in many plants, and exhibits antioxidant, anti-inflammatory, anticancer, and wound-healing properties. The main disadvantage related to the use of this molecule for pharmaceutical application is its poor bioavailability, due to its low solubility in aqueous media. Poloxamer 407-hydrogels show interesting thermo-sensitive properties that make them attractive candidates as pharmaceutical formulations. The hydrophobic domains in the chemical structure of the copolymer, a polymer made up of two or more monomer species, are useful for retaining poorly water-soluble compounds. In this investigation various poloxamer 407-based hydrogels containing rutin were developed and characterized as a function of the drug concentration. In detail, the Turbiscan stability index, the micro- and dynamic rheological profiles and in vitro drug release were investigated and discussed. Rutin (either as a free powder or solubilized in ethanol) did not modify the stability or the rheological properties of these poloxamer 407-based hydrogels. The drug leakage was constant and prolonged for up to 72 h. The formulations described are expected to represent suitable systems for the in situ application of the bioactive as a consequence of their peculiar versatility.

## 1. Introduction

Hydrogels are a peculiar class of drug-delivery systems characterized by a natural or semi-synthetic three-dimensional polymeric network [1]. Due to the presence of various hydrophilic functional groups in their backbone including carboxylic, amine, sulfate, and hydroxyl residues, they are able to retain considerable amounts of water and biological materials [2]. The properties of hydrogels—especially their porosity and the similarity they have to the extracellular tissue matrix—make them promising biocompatible drug-delivery devices [3].

Various classifications of hydrogels have been proposed [3]. They can be classified according to their source (natural, synthetic or hybrid), the nature of the employed polymers (homo-, co- or multipolymer derivatives), their physico-chemical structure (amorphous, semicrystalline and crystalline), and the responsiveness to physiological environment stimuli (e.g., pH, ionic strength or temperature) [1,3,4,5]. In particular, temperature-sensitive hydrogels are of great interest because their gelation can be triggered at body temperature [4,5]. That is, they are viscous liquids at room temperature allowing easy preparation and administration, and then become gels at the site of administration, offering a prolonged and controlled release of the drugs [6].

The application of hydrogels is often limited to the delivery of hydrophilic compounds [7]. In fact, their main disadvantage is their incompatibility with hydrophobic compounds due to their hydrophilic backbone, which limits drug loading as well as the homogeneous dispersion of poorly water-soluble derivatives within the hydrogel matrix [3,8]. In order to overcome this shortcoming, various strategies were adopted such as the nanoencapsulation of drugs within biocompatible delivery systems, the physico-chemical modification of the polymer and the use of organic solvents [9,10,11]. The integration of both organic and inorganic materials into the network of hydrogels is typically performed in order to improve their properties and functionalities, obtaining the so-called “hybrid materials” [12,13,14]. Hybrid materials are emerging as a promising class of drug-delivery systems thanks to the possibility of synergistically combining the advantages offered by different components into a single formulation [12]. Among these, several hydrogels based on double network technology have been proposed, which can be used for several applications [13,14].

Recently, hydrogels characterized with a network made up of small micelles were developed to improve their drug-loading capacity [8]. Many of the thermosensitive hydrogels are composed of amphiphilic block copolymers of poly(ethylene glycol) (PEG) and several hydrophobic polymer chains such as poly(propylene oxide), poly(D,L-lactide), poly(D,L-lactide-co-glycolide), poly(ε-caprolactone), poly[(R-3-hydroxybutyrate], polyphosphazenes, and polypeptides [3,11]. The presence of PEG provides water solubility, low immunogenicity and biocompatibility, while the hydrophobic units improve the loading capacity of poorly water-soluble molecules through micellization [8].

Poloxamers, also known by the names of Pluronic^®^, Synperonic^®^, and Lutrol^®^, are other examples of widely employed temperature-responsive polymers [15]. They are GRAS (generally recognized as safe) excipients belonging to the class of ABA-type triblock copolymers [16]. Their amphiphilic nature is due to the presence of a polyoxypropylene (POP) unit in their chemical structure which is linked to two polyoxyethylene (POE) chains, making them useful surfactants, stabilizers, and solubilizing and coating agents [17]. Among these, poloxamer 407 (P407) has been approved by the Food and Drug Administration (FDA) for human use and is widely employed in the biomedical field thanks to its low toxicity and compatibility with numerous excipients and biomolecules [18,19,20]. Poloxamer 407 (Appendix A) has been used as a detergent, surfactant, and stabilizer in various pharmaceutical formulations [9]. Recently, this material was successfully employed for 3D printing technology [21].

Their peculiar thermo-gelling property depends on their ability to self-assemble into micelles in an aqueous environment when they are above the critical micelle concentration (CMC). The micellization process is mainly a function of the hydrophobic portion: in fact, the CMC value is inversely proportional to the number of POP units present [22,23]. Moreover, the presence of hydrophobic domains helps to retain poorly water-soluble molecules in the hydrogels [24]. The solubilization of hydrophobic compounds in the micelle POP core is energetically favorable and does not require any energy input [11].

The purpose of this investigation was to evaluate the influence of rutin on the stability and rheological features of P407-based hydrogels in order to obtain suitable systems for the in situ administration of the active compound. Rutin (Appendix A), is a flavone glycoside characterized by a variety of biological and pharmacological activities, including anti-inflammatory and antioxidant properties, myocardial protection, antidiarrheal, antimutagenic and antitumor effects [25,26]. The main disadvantage related to the clinical application of rutin is its poor bioavailability, caused mainly by its poor water solubility [27]. Various strategies have been proposed in recent years for modulating the bioavailability of rutin such as the micronization of the powdered form of the drug, complexation with cyclodextrins and encapsulation into various micro- and nanocarriers (e.g., microemulsions, nanoemulsions, nanocrystals and nanosuspensions) [28,29,30,31,32].

Several authors have described the development of rutin-loaded hydrogels [33,34,35,36]. Tran and coworkers described a hydrogel made up of a rutin-chitosan derivative obtained by a synthetic approach for dermal wound repair [33]. In vitro and in vivo studies demonstrated that the system significantly enhanced cell proliferation and wound healing with respect to the commercial formulation Duoderm^®^ [33]. Another hydrogel formulation containing rutin was prepared using Carbopol Ultrez^®^ 10 NF and polysorbate 80 as its main components and proposed for skin application because of its interesting wound-healing properties [34].

Park et al. developed a system made up of rutin-loaded ceramide liposomes contained in cellulose-hydrogel and demonstrated that this formulation significantly increased the skin permeation of the drug with respect to the drug-loaded vesicular carriers or the gel matrix [35].

To the best of our knowledge, a formulation made up of P407-based hydrogels containing rutin has never been developed and characterized. For this reason, the aim of this investigation was to exploit the peculiar thermo-sensitive properties of this copolymer in order to obtain a conceivable novel formulation to be proposed for in situ administration of the drug. The influence of various concentrations of rutin (as a powder or solubilized in ethanol) on the physico-chemical and rheological features of P407-based hydrogels was investigated. In detail, the Turbiscan stability index (TSI), the micro- and dynamic rheological profiles, and the in vitro drug release profiles were evaluated and discussed (Figure 1).

## 2. Materials and Methods

### 2.1. Materials

Rutin hydrate, poloxamer 407 (P407, trademark Pluronic^®^ F127, molecular weight: 12,600 Da), and phosphate-buffered saline (PBS) tablets were purchased from Sigma Aldrich S.r.l. (Milan, Italy); cellulose membrane Spectra/Por molecular weight cut-off 3500 Da were obtained from Spectrum Laboratories Inc. (Eindhoven, The Netherlands). All other materials and solvents used in this investigation were of analytical grade (Carlo Erba, Milan, Italy). Deionized double-distilled water was used throughout the study.

### 2.2. Hydrogel Preparation

P407-based hydrogels were prepared using the “cold method” [37]. The composition of the various formulations is reported in Table 1. Briefly, an appropriate amount of P407 (20%, *w*/*w*) was slowly dispersed in distilled water under constant stirring at a controlled temperature of 4 °C and kept overnight until a clear solution was formed. In the same way, various amounts of rutin (as a powder or solubilized in ethanol) were added to the poloxamer “solution”. The obtained formulations were stored at 4 °C before the following analyses.

### 2.3. Stability Evaluation—Multiple Light Scattering

The formulations were analyzed by a Turbiscan^®^ Lab Expert apparatus (Formulaction, Toulouse, France) in order to evaluate their backscattering (Δ*BS*) and transmission (Δ*T*) profiles as a function of incubation time and temperature [38]. Briefly, 20 mL of each gel was placed into borosilicated cylindrical glass vials equipped with stoppers and scanned from bottom to top for a total of 1 h. The resulting data was processed by a Turby Soft 2.0 software program and reported as kinetic stability profile vs. time.

The TSI is a computation directly based on the raw data and sums up all the variations of BS and *T* signals in the sample. This statistical parameter is used to estimate the stability of samples and is calculated by the following equation:(1)$$\mathrm{TSI}=\frac{1}{N_{h}}\overset{t_{\max}}{\underset{t_{i}=1}{\sum }}\overset{z_{\max}}{\underset{z_{i}=z_{\min}}{\sum }}\left|{\mathrm{BST}}_{\left(t_{i,}z_{i}\right)}-{\mathrm{BST}}_{\left(t_{i-1,}z_{i}\right)}\right|$$
where *t*_max_ is the measurement point corresponding to the time t at which the TSI is calculated, *z*_min_ and *z*_max_ the lower and upper selected height limits respectively, *N_h_* = (*z*_max_ − *z*_min_)/Δ*h* the number of height positions in the selected zone of the scan and BST the considered signal (BS if *T* < 0.2%, *T* otherwise). [39]. The occurrence of sedimentation, creaming or flocculation was evaluated.

### 2.4. Rheological Analyses

#### 2.4.1. Diffusing Wave Spectroscopy

The microrheology of rutin-loaded P407 hydrogels was measured using a Rheolaser^®^ Master (Formulaction, Toulouse, France) the technology of which is based on multi-speckle diffusing wave spectroscopy (MS-DWS). An opaque sample is placed in a flat-bottomed cylindrical glass tube and illuminated by a laser beam (*λ* = 650 nm). The photons that penetrate the sample are backscattered due to the presence of particulate suspended in the formulation and an interference image called a “speckled image” is detected by a video camera. The Brownian movement of the particles in the sample causes the deformation of the speckle image and a detector records the dynamics of the deformation to quantify the speed of movement of the particles. A fast movement of the particles causes a fast deformation of the speckle image, while a slow motion of the particles leads to a slow deformation of the speckle image [40]. Standard numerical algorithms were used to quantitatively characterize the deformation rate of the speckle image and to obtain statistical parameters such as the decorrelation curve that characterizes the speed of the particles in the fluid and the mean square displacement (MSD) as a function of the decorrelation time and different kinetic parameters such as the elasticity index (EI) and the solid-liquid balance (SLB) [41]. The MSD value is determined from an autocorrelation function derived from the approach of Weitz and Pine and depends on the viscoelastic properties of the samples [40]. In a pure viscous fluid, the displacement of the particles increases linearly with time because they can move freely in the medium, while in a viscoelastic fluid, the particles are limited in their movement due to the polymeric network [39,40]. The MSD plot of a viscoelastic sample could be broken down into three regions with respect to the decorrelation time. At the very short decorrelation time, the particles are totally free to move in the medium, so the MSD curves increase linearly; at intermediate decorrelation times the particles interact with the polymeric “cage”, and the MSD slope decreases and reaches a plateau (this phase is related to the elasticity of the sample); finally, at longer decorrelation times, the particles escape from the “cage” and the MSD grows linearly again [39,40].

The elasticity index (EI) is the inverse of the MSD value and identifies the elasticity of a sample. It is computed from the plateau of the MSD curves at intermediate decorrelation timescales according to the equation:(2)$$\mathrm{EI}=\frac{1}{6\delta^{2}}\times \frac{d_{e}}{d}$$
where *6δ^2^* (nm^2^) is the mean value of MSD at intermediate decorrelation times, *d* (μm) is the mean particle diameter measured by laser diffraction, and *d_e_* is the diameter of a model particle used for calibration (1 μm) [40]. Lower MSD plateau values mean a smaller cage and, therefore, a stronger degree of elasticity of the systems [39,40].

The SLB (solid–liquid balance) provides information concerning the evolution of the ratio between the solid-like and the liquid-like behavior of the formulation as a function of time. The lower the value, the more “solid-like” the sample, the higher the value, the more “liquid-like” the sample [39]. It is simply the computation of the slope value of the MSD curve at the elastic plateau in logarithmic scale. In fact, if the slope is low and the movement of the particles is slow, then the sample has a behavior more similar to a solid than to a liquid; while if the slope is greater, the free movement of the particles indicates a more liquid sample [39,42].

The SLB can be expressed as a value between 0 and 1: a totally elastic system would typically have an SLB = 0. 0 < SLB < 0.5 means the sample is mainly solid-like, 0.5 < SLB < 1 shows a liquid-like behavior while a viscous formulation has an SLB = 1 [42].

The data was processed using RheoSoft Master^™^ 1.3.2.0 software (Formulaction, Toulouse, France) [40,43]. The hydrogels were enriched with latex beads (1 μm, 0.1 wt. %) and analyzed for 30 min at 37 °C.

#### 2.4.2. Dynamic Rheology

The dynamic rheological measurements were performed in triplicate using a Kinexus^®^ Pro rotational rheometer (Malvern Instruments Ltd., Worcestershire, UK) equipped with a Peltier element for temperature control, by employing a cone-plate geometry (diameter 40 mm; angle 2°). The gap between the plates was 1 mm. A solvent trap was attached to the rheometer to minimize the water evaporation that could occur during testing. Samples in the liquid state (1 mL) were loaded onto the lower plate using a spatula to ensure that formulation shearing did not occur [44]. Gelation rapidly took place on the rheometer plate. Before all rheological measurements, the samples were kept on the plate for 10 min, in order to facilitate the relaxation of any internal stress induced during loading [45]. Each sample was used for one test only. rSpace for Kinexus^®^ software was used to analyze the data.

The following tests were carried out:

**Shear viscosity measurements**: flow measurements were performed on the P407-hydrogels at 5 °C (below T_sol-gel_) and 37 °C (body temperature). The shear rates ranged from 1 up to 100 s^−1^.

**Oscillatory measurements**: the oscillatory measurements were carried out in order to investigate the elastic modulus (*G*′), the viscous modulus (*G*”), and the phase angle (*δ*). Initially, an amplitude sweep in the shear stress ranging from 0.1 to 100 Pa was performed at a fixed frequency of 1 Hz to define the linear viscoelastic region (LVR). The LVR is essential for the oscillation analysis because it is fundamental for preserving the internal structure of the sample during the measurements. A fixed stress amplitude of 1 Pa was chosen for the oscillatory measurements. In the frequency sweep experiments the samples were exposed to a step-wise process of increasing frequency (0.1–10 Hz) at 37 °C. Then, the *T*_sol-gel_ of the systems was investigated monitoring the variation of *G*’ and *G*” as a function of the temperature in the range from 10 to 40 °C (2 °C/min) at a fixed frequency value of 1 Hz. *T*_sol-gel_ was identified as the temperature at which the curves of the elastic and the viscous moduli cross each other, that is, when the samples exhibited a switch from a prevalently viscous behavior (*G*’’ > *G*’) to a prevalently elastic one (*G*’ > *G*’’) [16,46].

### 2.5. In Vitro Gel Dissolution Test

Gel dissolution studies were performed in triplicate and based on the gravimetric method [47,48]. 1 mL of liquid formulation (4 °C) was added to a pre-weighed glass vial and equilibrated at 37 °C for at least 10 min with the aim of promoting gelation. Afterward, the vials containing the samples were weighed and 2 mL of PBS (10 mM) previously equilibrated at 37 °C were added. At pre-determined incubation times, the supernatant was removed, the vials containing the remaining gels were re-weighed and fresh buffer added again. The weight of dissolved hydrogels was calculated from the difference in weight of the vials.

### 2.6. Evaluation of the Drug-Release Profiles

The release profiles of rutin from the P407-based hydrogels was evaluated using the dialysis method by means of cellulose acetate dialysis tubing with a molecular cut-off of 3500 Da, sealed at both ends with clips [46]. PBS (10 mM) was used as the release medium, which was constantly stirred and warmed (GR 150 thermostat, Grant Instruments Ltd., Cambridge, UK) to 37 ± 0.1 °C throughout the experiment. 4 mL of each formulation in the liquid state were placed in the dialysis bag which was then transferred to a beaker containing 100 mL of the release medium in order to operate under sink conditions for 72 h; 1 mL of release medium was withdrawn and replaced with fresh medium at various incubation times.

The samples were analyzed at *λ*_max_ 362 nm of rutin by a PerkinElmer Lambda 35 ultraviolet–visible (UV–vis) spectrophotometer equipped with PerkinElmer acquisition software (Perkin-Elmer GmbH Uberlingen, Germany). The following calibration curve of rutin was used:*y* = 0.018*x* + 0.0005,
where *y* is the drug concentration (µg/mL) and *x* is the absorbance at 362 nm; the *R*^2^ value was 0.999.

The drug release studies were performed in triplicate [25].

### 2.7. Statistical Analysis

Statistical analysis of the various experiments was performed by analysis of variance (ANOVA) with a *p* value of <0.05 considered statistically significant.

## 3. Results and Discussion

### 3.1. Preparation and Stability Evaluation of P407-Based Hydrogels

Despite the numerous valuable properties of hydrogels, their use has generally been limited to the delivery of hydrophilic drugs [3]. The presence of the POP hydrophobic domain in the structure of P407 promotes interaction with lipophilic drugs and their retention in the copolymer-based hydrogels [24,49,50,51,52].

Moreover, P407 aqueous “solutions” show interesting thermo-sensitive properties useful for the development of various pharmaceutical formulations [9]. Previous investigations demonstrated that hydrogels made up of 20% (*w*/*w*) of P407 are characterized by an ideal gelation temperature (20–25 °C) and noteworthy manageability [53,54,55,56].

In this investigation, various P407 (20%, *w*/*w*)-based hydrogels containing different amounts of rutin were prepared in order to identify the saturation limit of the poloxamer “solution” (Table 1).

Rutin was added in powder form or solubilized in ethanol up to a concentration of 0.1% (*w*/*w*). Greater amounts of the drug elicited the appearance of drug precipitation or macro-aggregation phenomena (Appendix A).

It is well known that ethanol influences the gelling properties of poloxamer-aqueous systems [57,58] so the effect of this solvent (2% *w*/*w*) on the stability and rheological properties of the polymer dispersion was investigated. The amount of ethanol was chosen considering the minimum volume necessary to solubilize the rutin at the highest concentration used.

The stability of P407-based hydrogels containing rutin at different concentrations, either with or without ethanol, was assessed by multiple light scattering (MLS) using a Turbiscan^®^ Lab Expert (Formulaction, France). The analyses were carried out at 4 °C (storage temperature), 25 °C (room temperature), and 37 °C (body temperature) and reported as TSI variation. As shown in Figure 2, the stability kinetic profiles of the various formulations are very similar and no significant variation resulted over time, evidencing the absence of any phenomena such as sedimentation, migration and/or flocculation, as was also demonstrated by the ∆*BS* and ∆*T* (Appendix A).

The presence of rutin and ethanol in the P407-based hydrogels did not cause significant variation of the stability of the systems at any of the investigated temperatures. In particular, the systems containing rutin at a drug concentration of less than 0.1% (*w*/*w*) showed no significant variations of TSI with respect to the empty formulations. On the contrary, amounts of the active compound above 0.1% *w*/*w* showed sedimentation and the formation of macroaggregates at 4 °C (Appendix A).

### 3.2. Rheological Evaluation

#### 3.2.1. Diffusing Wave Spectroscopy

Rheology is the study of the deformation and the flow characteristics of materials subjected to an external force or stress [59]. In micro-rheology, the particles contained in a formulation are used to measure the local deformations that occur due to an applied external stress (active micro-rheology) or simply through the evaluation of their Brownian movement (passive micro-rheology) [60]. The Rheolaser^®^ Master uses the MS-DWS technique for the micro-rheological analyses of viscoelastic systems at rest [42]. Figure 3 shows the MSD curves of the P407-hydrogels both as empty formulations and prepared with ethanol (2%, *w*/*w*).

The instrument can control temperatures between room temperature and 90 °C. For this reason, the authors analyzed the formulations taken directly from the storage temperature (4 °C)—that is, in a liquid stage—in order to evaluate their phase transition when incubated at 37 °C.

In the initial stage of the test the formulations had a liquid-like behavior and the slope of the MSD curves (blue lines) is characterized by a value greater than that of the final phases of the same analysis (red lines), demonstrating a decrease in the movement of the particles (Figure 3). This was due to the gelation process occurring as the temperature increased, confirming the characteristics of thermo-sensitivity of the formulations and the sol-gel transition of the polymeric network. The same trend was observed in the formulations containing various amounts of rutin added as a powder or solubilized in ethanol during the preparation procedure (Appendix A). These results demonstrated that the presence of the drug and the organic solvent does not significantly modulate the thermo-sensitive behavior of P407-hydrogels.

The transition from the liquid-like to the solid-like phase was confirmed by the SLB (solid–liquid balance) values of the formulations extrapolated from the MSD curves. In fact, the MSD curves can be considered the viscoelastic fingerprint of the analyzed formulations and can be used to obtain other kinetic parameters such as the elasticity index (EI) and the SLB [42,43,60,61,62,63].

Figure 4 shows the SLB values of the samples at the beginning (*t_i_*) and at the end of the analysis (*t_f_*). The specific SLB values are reported in Appendix A.

The results confirm the behavior of the formulations previously described. In fact, the SLB values of all the formulations have been shown at the beginning (*t_i_* = 1 min) and at the end (*t_f_* = 30 min) of the analysis with the aim of demonstrating that rutin does not dramatically modify the typical thermo-sensitive behavior of the poloxamer-based hydrogels. In all cases, the formulations showed an SLB value greater than 0.5 in the first minute of analysis, confirming their liquid behavior, while it was close to 0 by the end of the analysis, showing the phase transition of the formulations at 37 °C.

Figure 5 shows the EI profiles of the various formulations as a function of time, confirming that the presence of the drug and the organic solvent induced no variation in the microrheological behavior of the systems.

#### 3.2.2. Dynamic Rheological Measurements

Rheological analysis is an important method for characterizing the semi-solid dosage formulations because it can provide useful information on their in vivo behavior. In fact, it is well known that the flow characteristics of the formulations influence the residence time at the administration site as well as the release rate of the retained active compounds [44,64].

The mechanical properties of poloxamer hydrogels can be affected by the physico-chemical features of the entrapped drugs and/or additives [65,66], so the effect of rutin and ethanol on the rheological properties of P407-based hydrogels was investigated. Co-solvents such as ethanol or other organic solvents are usually employed to entrap poorly water soluble active pharmaceutical ingredients into poloxamer solutions [2,45].

The dynamic rheological characterization of P407-based hydrogels was performed at 4 °C and at 37 °C to simulate the storage and in vivo conditions. The shear viscosity profiles of the different formulations are shown in Figure 6 and Figure 7. The results show no significant difference in viscosity with respect to the that of the drug—and ethanol—free samples. The obtained results are in agreement with those described by Fakhari et al. who evaluated the effect that the addition of ethanol at different concentrations has on the gel formation, strength and the viscosity of “P407 solutions”. In fact, only high ethanol concentrations favored a gradual decrease of the gelation temperature and an increase of viscosity [66].

All investigated systems exhibit a Newtonian and non-Newtonian shear thinning behavior at 4 °C and 37 °C, respectively, in agreement with previous experimental investigations [66,67,68,69]. In fact, the temperature and the polymer concentration can influence the rheological features of P407 “solutions” which can exhibit either Newtonian (below the *T*_sol-gel_ and CMC) or non-Newtonian (above the *T*_sol-gel_) behavior [9,16,67].

The viscosity *η* of the P407-based hydrogels decreased when the shear rate *γ* increased, a trend which promotes the flow of the formulations. The shear thinning behavior is preferred in the design of an in situ gel system, because the formulation can be easily injected using minimal pressure [70]. Shear-thinning properties make the delivery of the formulation through a syringe and needle easier because the increase of the shear rate in the needle favors a decrease in the apparent viscosity, due to a temporary destructuration of the polymeric network [71].

In order to investigate the influence of rutin on the viscoelastic properties of the formulations, the mechanical behavior of the P407-based hydrogels containing various amounts of rutin was evaluated.

Oscillatory studies determine the viscoelastic properties of the formulations by means of the application of a sinusoidal shear stress. The principal parameters obtained are the elastic modulus *G*’, the viscous modulus *G*” and the phase angle (*δ*). *G*’ describes the ability of the sample to store elastic energy (storage modulus), while *G*” defines the ability to dissipate it (loss modulus) [16]; the phase angle provides a measure of the viscoelastic balance of the behavior of the material. If the *δ* value is 90° the material can be considered to be purely viscous, whereas if the *δ* value is 0°, this describes a purely elastic material [72]. These parameters are used to ascertain whether a formulation can be defined as a gel, i.e., a system where *G*′ and *G*” are frequency-independent and the phase angle δ is low at all the tested frequencies [73].

The LVR of the formulations was determined by the evaluation of *G′* and *G*” as a function of a shear stress amplitude sweep (data not shown). The end of LVR is the point where an alteration of the internal structure of the hydrogel occurs. For this reason, the frequency sweep experiments were performed at a constant shear stress within the LVR with the aim of preserving the internal structure of the systems during the experiment [74]. *G*′ and *G*” of P407-hydrogels are reported in Figure 8 as a function of frequency. All samples showed pronounced, gel-like behavior with the storage modulus (*G*′) much higher than the loss modulus (*G*”), and very low phase angle values.

The presence of rutin within the polymeric matrix did not modify the viscoelastic behavior of the system, which still appeared typical of a gel-like material (Figure 8). Moreover, as expected, the profiles of the formulations containing rutin added as a free powder or previously solubilized in ethanol are superimposable. These profiles are very similar to that of the rutin-and ethanol-free formulation (Appendix A). The values of the phase angle (*δ*) of the various hydrogels are reported in Appendix A.

The *T*_sol-gel_ of the polymeric systems was investigated by means of the oscillatory measurement test, monitoring the variation of the elastic and viscous moduli as a function of the temperature in the range between 10 and 40 °C. *T*_sol-gel_ was identified as the temperature where *G*’ and *G*’’ intersect, that is, the temperature at which the sample exhibits a switch from a prevalently viscous behavior (*G*’’ > *G*’) to a prevalently elastic one (*G*’ > *G*’’) [16]. The results are reported in Table 2; the *T*_sol-gel_ value of the various formulations was approximately 24 °C, typical of a P407-hydrogel prepared with 20% *w*/*w* of the copolymer [75,76].

The phenomenon of thermo-gelation depends on the hydrophobic interaction between the copolymer chains [77,78]. In fact, upon increasing the temperature, the P407 blocks start to aggregate in micelles due to the dehydration of the POP units. This is the first step of the gelation process [16,66].

This characteristic was confirmed by the temperature sweep measurements performed on the various formulations as a function of the temperature (Appendix A). In fact, in the first step of the analysis, the viscous modulus was much higher than the elastic one and the phase angle was close to 90° (<*T*_sol-gel_); when the *T*_sol-gel_ is reached, the profile is reversed as a consequence of the gelation that occurs (Appendix A).

Appendix A shows the effect of temperature on the complex modulus (*G**) for all formulations. At temperatures below the *T*_sol-gel_, the values of *G** were low, indicating the liquid-like behavior of systems; a dramatic increase of *G** occurred above 24 °C, demonstrating the gelation of the formulations. All the analyzed samples showed very similar sol-gel rheological profiles, confirming the previously obtained results that are very similar to those reported in literature when the same concentration of P407 is used to obtain hydrogels [79,80,81].

### 3.3. In Vitro Dissolution and Drug-Release Profiles

The dissolution profile of P407-hydrogels should be evaluated because of their conceivable in situ application [76,82]. The release of the entrapped bioactive compound(s) occurs through gel erosion. This process can be influenced by the copolymer concentration, by the presence of additives (such as co-solvents or bioadhesive polymers) and by the physico-chemical characteristics of the drugs.

The dissolution profile of the empty formulation is shown in Figure 9. As previously confirmed by other authors, the dissolution of the hydrogel occurs constantly and rapidly, that is, virtually 100% after 24 h [48]. This result is probably due to the high solubility of P407 in aqueous media and this is important for the design of an in situ gelling formulation because the dissolution properties modulate the drug leakage at the administration site [83]. Again, neither the rutin nor the ethanol influenced the dissolution rate of the P407-hydrogels (Figure 9 and Figure 10).

Unlike covalently cross-linked gels which are able to promote a prolonged release of the entrapped compounds, poloxamer-based systems are normally characterized by a faster drug leakage [84].

The amount of rutin released from the P407-based hydrogels was evaluated at 37 °C using the dialysis method and the results are reported in Figure 11. When P407 hydrogels are separated from the release medium by a dialysis membrane, the dissolution of the hydrogel is almost prevented, and the diffusion of the drug from the gel matrix dominates the release mechanism of the active compound [9,84].

The drug leakage was constant and prolonged for up to 72 h for all the investigated formulations; the amount of rutin released in the first few hours was similar among the various systems and only a small increase in the release rate of rutin was observed after 24 h when ethanol was used, probably as a consequence of the modulation of the microenvironmental properties exerted by the organic solvent, which promotes greater leakage of the active compound from both the polymeric network of hydrogels and the P407 micelles [58].

The same experiment was performed at storage temperature (6–8 ° C, in a cold room) in order to investigate the release profile of rutin from the liquid-like systems (Appendix A). The drug was released quicker with respect to the solid-like systems (especially during the first few hours) even though the leakage of rutin remains gradual over time and confirmed the role of both P407 micelles and the gel network in the modulation of this parameter. Also in this case, ethanol promoted an increased leakage of rutin confirming its influence on the different localization of the active compound in the system (P407 micelles/water) as previously discussed for the SLB investigation.

## 4. Conclusions

The entrapment of a bioactive compound in a thermo-sensitive gelling system is a potential approach to be used for in situ administration due to the easy injection of the formulation and a controlled release of the retained drug.

In this paper, we have shown for the first time the feasibility of using P407-based hydrogels for the in situ administration of rutin. The entrapment of the drug within P407-based hydrogels is expected to have many advantages: their liquid-like behavior at low temperature values allows an easy preparation procedure and manageability and makes them suitable injectable formulations. The gelation that comes about in the body could prolong their residence time at the application site, and enable a controlled release of the entrapped drug, decreasing its potential side effects and the number of necessary in vivo administrations. These features could promote greater patient compliance and decrease the costs of therapy.

The addition of rutin to P407 solutions (20%, *w*/*w*) did not compromise the physico-chemical properties of the resulting hydrogels up to a drug concentration of 0.1% *w*/*w*. Moreover, the rheological profiles of systems prepared with rutin as a free powder were comparable to those obtained using an ethanol solution of the active compound; this could mean the possible administration of a drug without the use of co-solvents, avoiding potential adverse effects [85,86].

The ability of P407 to promote the solubilization of poorly water-soluble molecules is a well-known characteristic of this material [9,50,87]. Rutin can efficiently interact with the copolymer (micelles and gel network) as a consequence of hydrophobic as well as electrostatic interactions due to its peculiar chemical nature [88,89].

Thanks to their versatility, these formulations could be employed for various biomedical applications and administration routes (e.g., subcutaneous or intramuscular) [45,48,84,86]. For examples, previous studies demonstrated that rutin significantly increases the fibroblast proliferation and the collagen production, being a support in wound-healing therapy [33]. P407 has been successfully used for the treatment of burn wounds [37,90,91]. In addition, the therapeutic anti-inflammatory or antioxidant properties of the bioactive contained in P407 hydrogels may be improved as was true in the case of other drugs administered as poloxamer-based formulations [16,92,93,94,95].

Consequently, additional in vivo studies are in progress in order to evaluate the real effectiveness of the described formulations, the dose and time-dependent toxicity, the in situ residence time of the systems and the pharmacokinetic profile of rutin.

## Figures and Tables

**Figure 1 nanomaterials-10-01069-f001:**
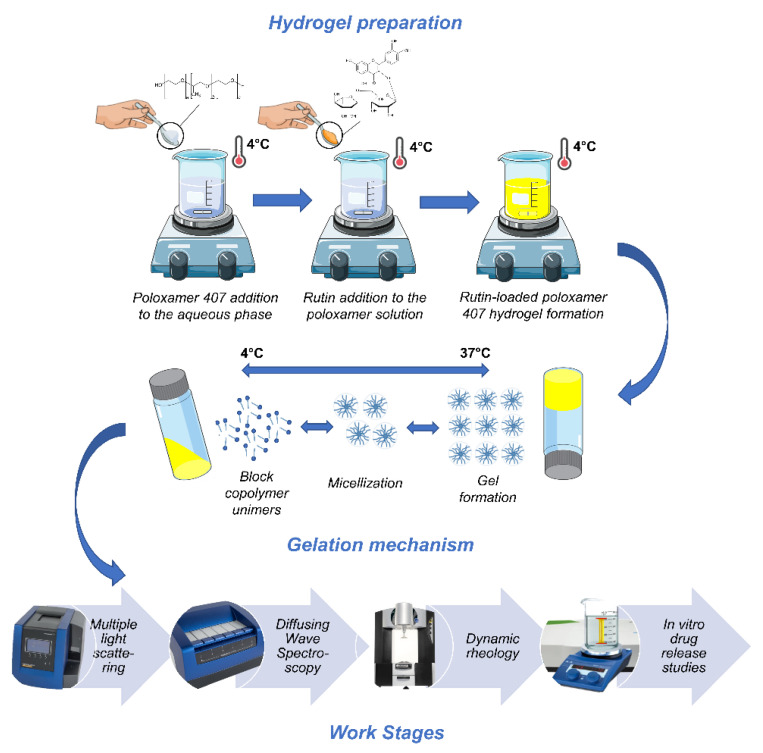
Schematic representation of the work stages.

**Figure 2 nanomaterials-10-01069-f002:**
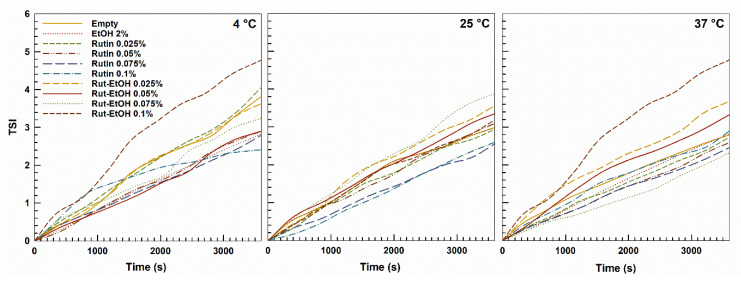
Turbiscan stability index (TSI) of P407 (20%, *w*/*w*)-based hydrogels with and without ethanol (2%, *w*/*w*) as empty formulations or containing rutin as a function of temperature and incubation time. The result was a representative experiment of three independent experiments.

**Figure 3 nanomaterials-10-01069-f003:**
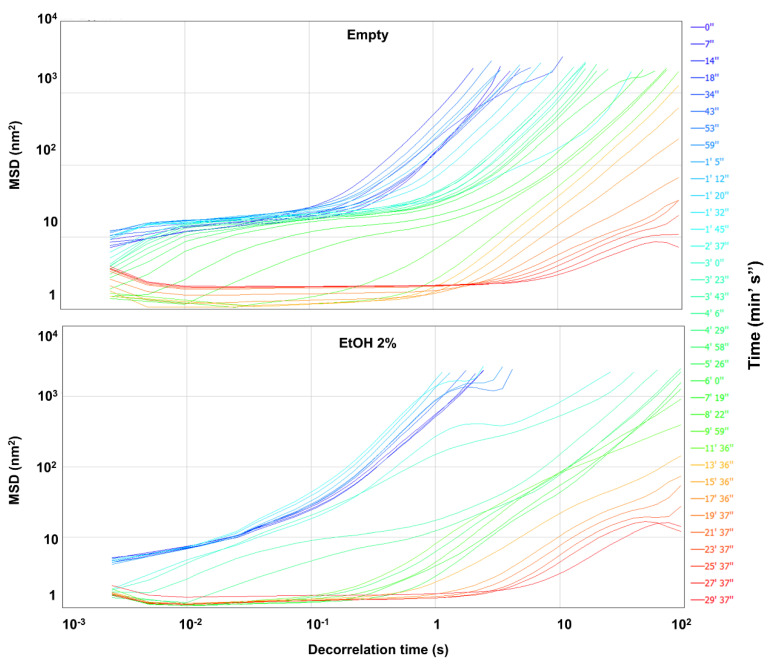
Mean square displacement (MSD) curves of P407 (20%, *w*/*w*)-based hydrogels as a function of the decorrelation time. The different colors of the curves correspond to a single scan performed throughout the analysis. The analysis was performed in the range from 4 to 37 °C.

**Figure 4 nanomaterials-10-01069-f004:**
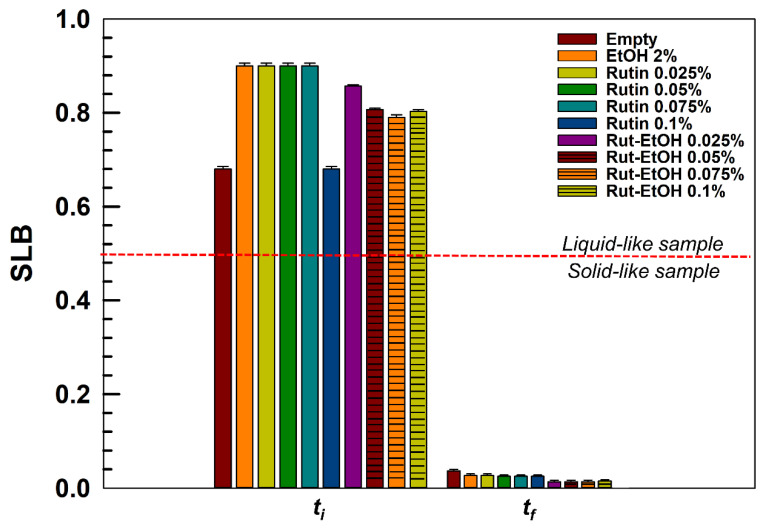
Values of SLB (solid–liquid balance) of the of the various P407-based hydrogels at the beginning (*t_i_* = 1 min) and at the end (*t_f_* = 30 min) of the analysis.

**Figure 5 nanomaterials-10-01069-f005:**
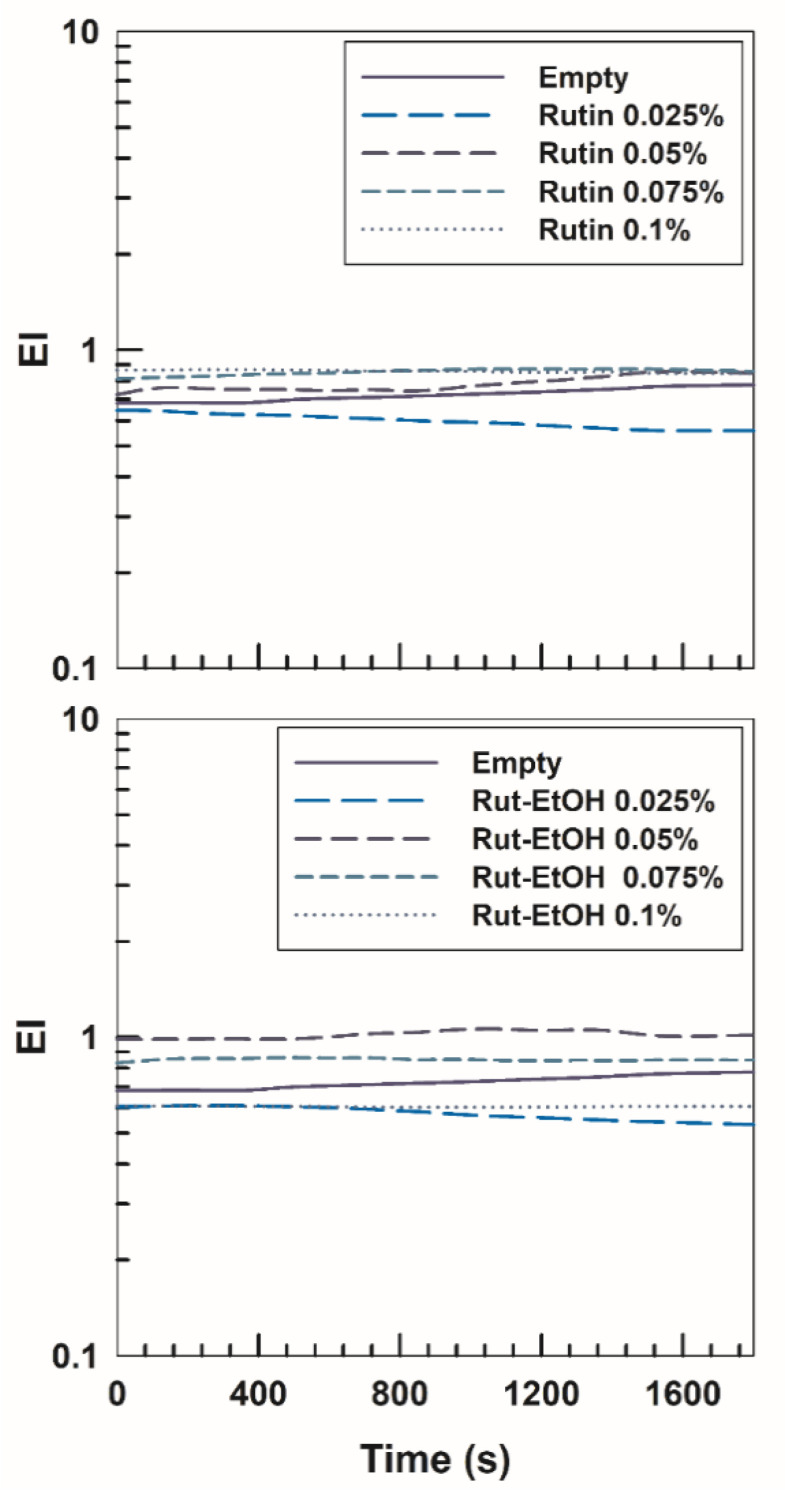
Elasticity index (EI) of rutin-loaded P407 hydrogels as a function of the time. The analysis was performed at 37 °C.

**Figure 6 nanomaterials-10-01069-f006:**
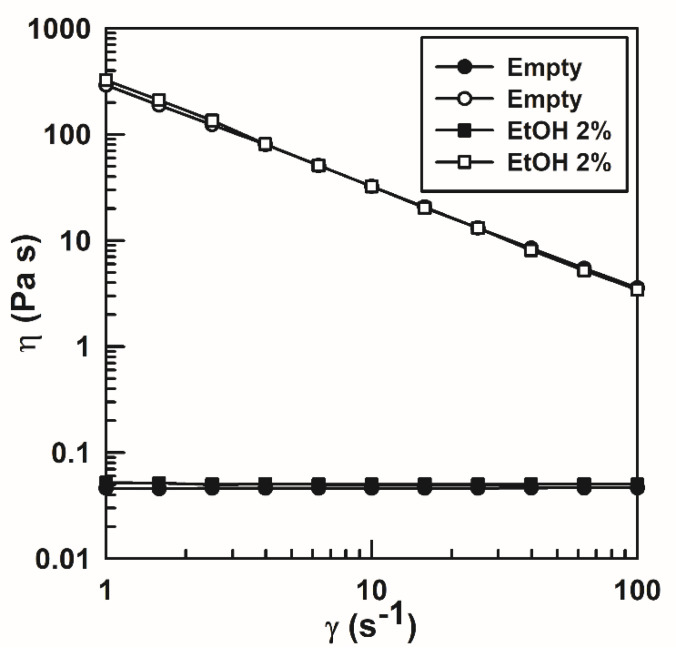
Viscosity profiles of P407 (20%, *w*/*w*) hydrogels as an empty formulation or prepared using ethanol (2%, *w*/*w*) as a function of the shear rate. The analysis was performed at 4 °C (black symbols) and 37 °C (white symbols).

**Figure 7 nanomaterials-10-01069-f007:**
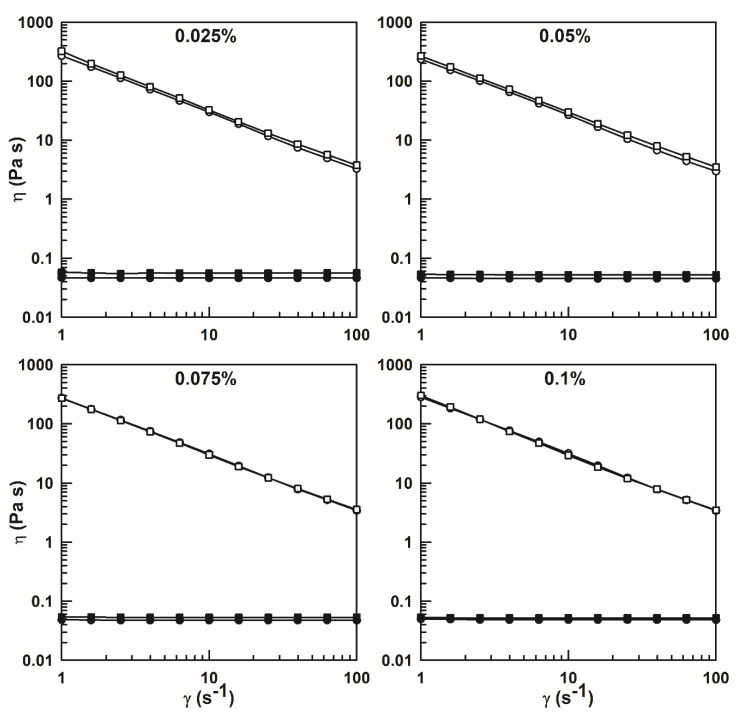
Shear rate viscosity of P407 (20%, *w*/*w*) hydrogels containing rutin added as a powder (circles) or previously solubilized in ethanol (squares) at 4 °C (black symbols) and 37 °C (white symbols).

**Figure 8 nanomaterials-10-01069-f008:**
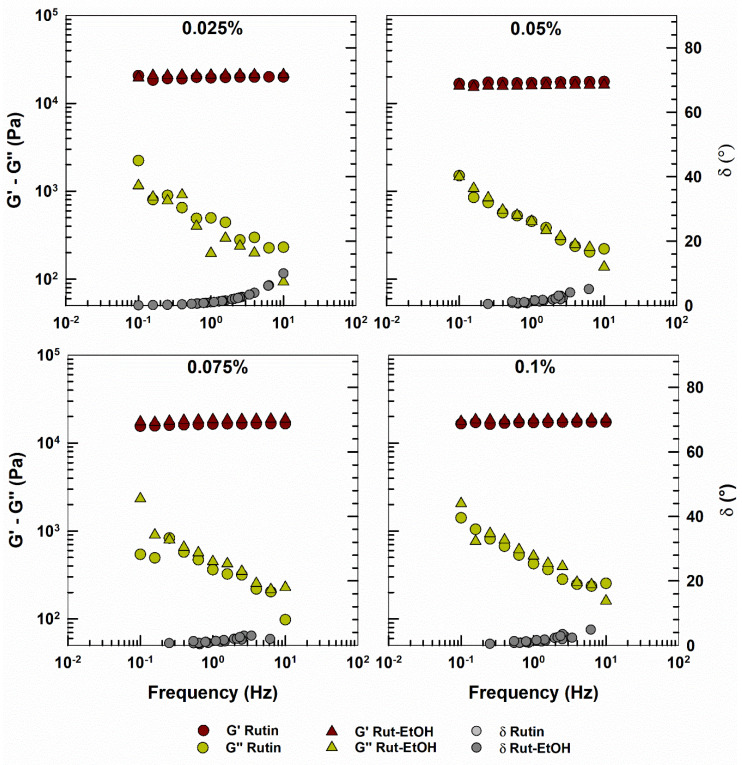
Elastic modulus (*G*’), viscous modulus (*G*’’) and phase angle (*δ*) of rutin-loaded P407 hydrogels as a function of frequency. The analysis was performed at 37 °C.

**Figure 9 nanomaterials-10-01069-f009:**
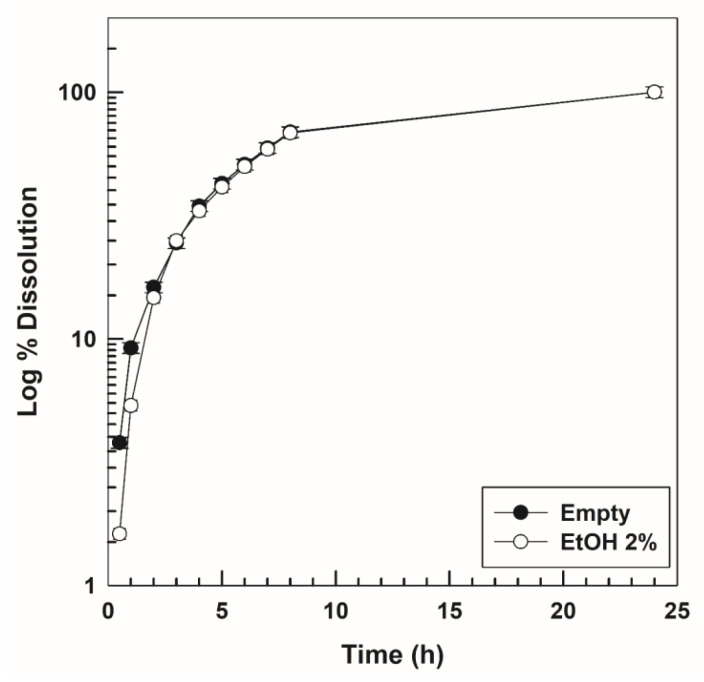
Dissolution profiles of P407 hydrogels as an ethanol-free system or containing the organic solvent (2%, *w*/*w*) as a function of the incubation time. Results are the mean of three different experiments ± standard deviation.

**Figure 10 nanomaterials-10-01069-f010:**
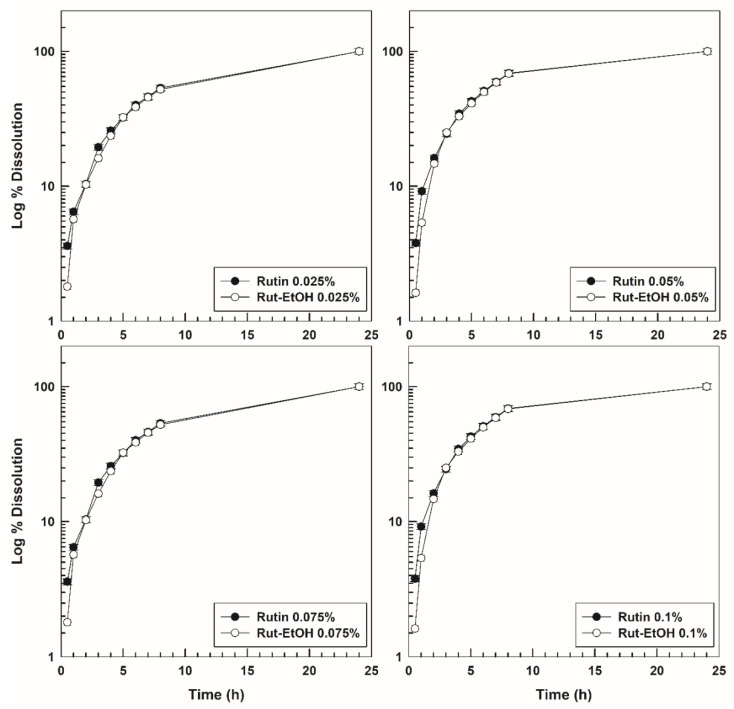
Dissolution profiles of various rutin-loaded P407 hydrogels as a function of the drug concentration and the incubation time. Results are the mean of three different experiments ± standard deviation.

**Figure 11 nanomaterials-10-01069-f011:**
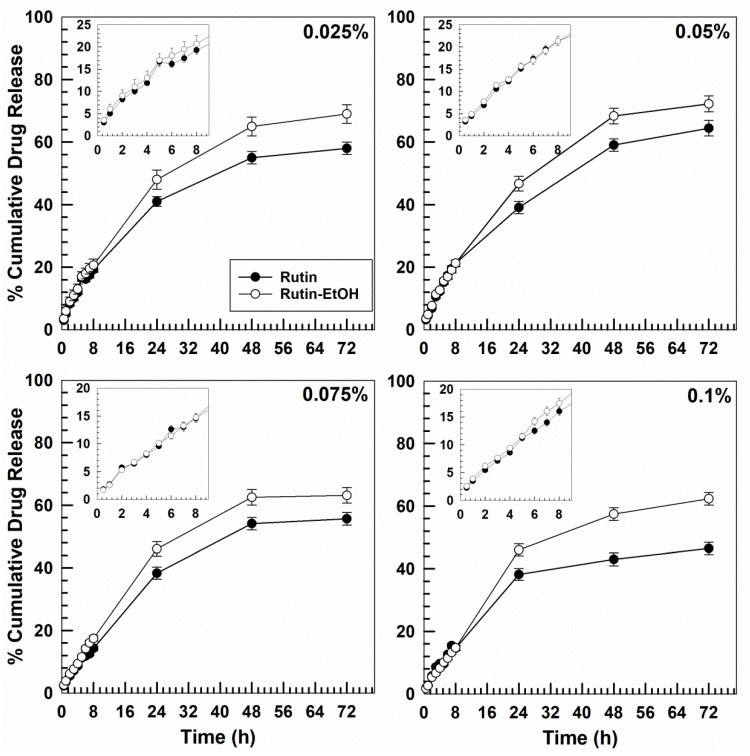
Release profiles of rutin in phosphate-buffered saline (PBS) from P407-based hydrogels. Experiments were carried out at 37 °C. Values are the average of three different experiments ± standard deviation.

**Table 1 nanomaterials-10-01069-t001:** Composition of various P407 hydrogels (20% *w*/*w* of copolymer).

Formulations	Rutin (%, *w*/*w*)	Ethanol (%, *w*/*w*)
Empty	-	-
EtOH	-	2
Rutin	0.025	-
	0.05	-
	0.075	-
	0.1	-
Rut-EtOH	0.025	2
	0.05	2
	0.075	2
	0.1	2

**Table 2 nanomaterials-10-01069-t002:** *T*_sol-gel_^1^ of the P407-based formulations.

Formulations	Rutin (%, *w*/*w*)	*T*sol-gel ^1^ ( °C)
Empty	-	24.08
EtOH 2%	-	24.08
Rutin	0.025	23.99
	0.05	24.05
	0.075	24.02
	0.1	24.05
Rut-EtOH	0.025	24.03
	0.05	24.02
	0.075	24.02
	0.1	24.10

^1^ Solution-gel transition temperature.

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
