# Peer review of "Rutin-Loaded Poloxamer 407-Based Hydrogels for In Situ Administration: Stability Profiles and Rheological Properties"

_nanomaterials, 2020, doi:10.3390/nano10061069_

Round 1

Reviewer 1 Report

The topic is up to date and offers inovative solutions. The article is well written and the results achieved are clearly presented and documented by a large number of graphs.

In the introduction, the section on the application of hydrogels for the delivery of hydrophobic drugs should be extended to other options (e.g. use of polyethylene glycols).

Author Response

Reviewer 1: The topic is up to date and offers innovative solutions. The article is well written and the results achieved are clearly presented and documented by a large number of graphs.

In the introduction, the section on the application of hydrogels for the delivery of hydrophobic drugs should be extended to other options (e.g. use of polyethylene glycols).

Authors: The authors thank the Reviewer for the positive judgment. In accordance with the Reviewer’s suggestion, the introduction has been duly revised (lines 48-55).

Reviewer 2 Report

The authors investigated rheological properties and dispersion stability of poloxamer hydrogels containing rutin. Although this study may be designed for any specific clinical application, the obtained results are straightforward but not surprising, and therefore a novelty of the study seems not very clearly presented. Specific comments to the authors are listed below.

(1) Please define the elasticity index (EI) and the solid-liquid balance (SLB) so that readers can see how these parameters are calculated from MSD. Similarly, please show the definition of Turbiscan Stability Index (TSI).

(2) Please show the reason why TSI increased with time in Fig. 1.

Reviewer 3 Report

Review: nanomaterials-752581
Title: Rutin-loaded poloxamer 407-based hydrogels: stability profiles and rheological properties.
Authors: Giuliano (first), Cosco (last)

Summary: This study examines the ability to trap the hydrophobic drug Rutin into the P407 nanostructure with/without EtOH as a cosolvent. Turbidity measurements are used to determine stability. Although it is unclear what the authors mean by stability (see below). The authors use an indexer to measure the quiescent Solid-liquid Balance (SLB) and Elasticity Index (EI) via light scattering methods. The SLB is a qualitative index between 0 and 1 that is related to the viscoelastic properties of the gel. EI is the inverse of the slope of MSD and is related to the elasticity of the sample. The authors also perform standard rheological measurements as a function of frequency and temperature. The presence of drug and drug+EtOH do not seem to affect the P407 phase transition, although some questions remain regarding the choice of parameters used to compare the various formulations (See below). The different formulations are studied in %dissolution and %drug release. In both studies, the authors found that neither the dissolution rate nor the drug release kinetics were affected by the presence of drug or drug+EtOH. Although the amount of drug released in the case of drug+EtOH is more than in the pure Rutin case. Unfortunately there was no comparison of the drug release kinetics to pure Rutin in the F127(P407) case below the gel transition temperature. This baseline would have significantly improved the context of drug release rates due to the gel state.

Review: The author added exceptionally small amounts of drug and EtOH. The reviewer did not expect a change in rheology and no change was reported. The results seem very expected. It could be that this concentration of Rutin is what is clinically relevant, however this was not made clear by the authors. Furthermore, if this is not a clinically relevant concentration, then this significantly reduces the impact of the work. If this is clinically relevant, how much of the formulation would need to be delivered to the patient, how would it be administered (subcutaneous, intravenous, etc). Lastly, it is not clear that the data supports the conclusions since the chosen parameters of comparison may not be the most relevant (especially in the temperature sweep rheology). The reviewer hopes that these comments will help to strengthen some of the arguments and looks forward to a revised manuscript. The authors are missing significant references regarding the rheological study of pluronic systems. Groups to look at are Lynn M. Walker, Carnegie Mellon University, for example: Rheology and phase behavior of copolymer-templated nanocomposite materials DC Pozzo, KR Hollabaugh, LM Walker - Journal of Rheology, 2005 - https://doi.org/10.1122/1.1888665 . This is only one example.

(i) Why does the "empty" P407 solution show increase in turbidity below the gel transition temperature. In Figure 1, we see that the turbidity increases with time at 4oC. This is unclear. The P407 solution should be remain clear and infinitely stable at 4oC. Please explain. The reviewer appreciates that because there are no differences in the TSI with th different formulations that they all exhibit the same stability, but what is not clear is why TSI is increasing with time in the first place. Overall it is not clear what is the significance of the increasing TSI value with time. NOTE: The colors and symbols are different than legend in Figure 1. There is two black (or brown) solid lines and no mustard color solid line.

(ii) What is the size of the Rutin in the formulations. In the case of EtOH, the Rutin is molecularly dissolved. In the pure P407 case, is the Rutin dissolved or is it forming small aggregates that are kinetically trapped by the

(iii) What temperature does table 2 represent? Although temperature is changing during this experiment, it would be interesting to estimate the temperature range that the SLB is being measured, i.e. over what range of temperatures is the slope evaluated?

The authors state that the SLB goes from 0.5 to 0 by the end of the experiment. Table 2 does not have any values for the final time analysis, despite having a footnote referring to the final value?

(iv) Why do the authors choose not to discuss the effect observed at Rutin 0.1%. Why does increasing the concentration of Rutin cause the system to behave (SLB value) as the "empty" system? More importantly, why does a small addition of Rutin (0.025%) change the SLB value by so much? Having all the data in the SI is a little annoying. It might be better to take a representative curve at short time and long time for each sample and compare.

(v) The meaning of the colors and the right axis in Figure 2 is unclear. more explanation is needed.

(vi) Figure 6 should be represented on a log-log plot for modulus and frequency. tan delta should be log (frequency) -linear (tan delta).

(vii) A linear-viscoelastic temperature sweep is more interesting to compare than the final gel state, as it details the transition from sol-gel. This data is currently in the supplementary. Surprisingly the crossover temperature as reported in Table 3 does not depend on the presence of Rutin or ethanol. It has been shown previously that the crossover is not a reliable measure of the sol-gel transition. Instead, it is best to use the maximum in dG*/dT. Is the maximum in dG*/dT affected by the presence of Rutin. It seems that there are subtle changes in the slop of G* with T, which seems to suggest that there would be. I acknowledge that the authors may not have conducted small enough step changes in temperature to do this analysis reliably.

(viii) The authors clearly show in Figure 8 that the gel is completely dissolved after 24 hours, but most of the gel is dissolved after 10 hours. The diffusion studies presented in Figure 9 show release kinetics much slower than these timescales. This of course means I am not able to correlate the diffusion studies to the relevant drug application physics. It would have been useful if the authors had conducted a control diffusion study, whereby Rutin was placed into the dialysis membrane with poloxamer, but kept below the transition temperature for the duration of the release study. This would at least shown the release kinetics in the absence of gel transport and melting physics.

(ix) It is not clear, nor is it discussed, why the EtOH increases the overall % cumulative drug release. The kinetics (rates of release) seemed to be unaffected by EtOH, which is to be expected since the rheology is unchanged, but it is not clear why less drug is released in the pure Rutin case.

(x) It would also be effective to summarize the initial kinetic drug release rates, i.e. the slope of the %drug release at early time to definitively show that the rates are unaffected. They look the same, but because of the large data range, it is unclear.

(xi) The authors do not discuss the solubility of the rutin in pure water as compared to the P407 solutions. Does the presence of P407 increase solubility?

minor comments/questions:

Line 111, are the vials sealed during the measurement?

Line 144, what temperature was the sample loaded?

Line 80 is missing citation (Park et. al.)

Line 81 missing comma "To the best of our knowledge, a"
